# MicroRNA Let-7 Plays an Important Role in the Immunopathology of COVID-19: A Systematic Review

**Renato Luís Pessôa** [1] **, Gustavo da Rosa Abreu** [1] **and Ramatis Birnfeld de Oliveira** [2,*]

1    Medical Sciences, School of Medicine, University of Vale do Taquari-Univates, Lajeado CEP 95914-014, RS, Brazil
2    Health and Medical Research Group, Medical Sciences, School of Medicine, University of Vale do Taquari-Univates, Lajeado CEP 95914-014, RS, Brazil
*    Correspondence: ramatisdeoliveira@gmail.com; Tel.: +55-51-37147000

**Abstract:** COVID-19 has presented itself as a challenging task to medical teams and researchers throughout the world, since the outbreak of SARS-CoV-2 started in the Chinese city of Wuhan. To this day, there are still new variants emerging, and the knowledge about the mechanisms used by the virus to infect cells and perpetuate itself are still not well understood. The scientific community is still trying to catch up with the velocity of new variants and, consequently, the new physiological pathways that appear along with it. It is known that the new coronavirus plays a role in changing many molecular pathways to take control of the infected cells. Many of these pathways are related to control genomic expression of certain genes by epigenetic ways, allowing the virus to modulate immune responses and cytokines production. The let-7 family of microRNAs, for instance, are known to promote increased viral fusion in the target cell through a mechanism involving the transmembrane serine protease 2 (TMPRSS2). It was also demonstrated they are able to increase the inflammatory activity through the NF-κB/IL-6/let-7/LIN-28 axis. In addition, let-7 overexpression led to a reduction in inflammatory cytokines and chemokines expression (IL-6, IL-8 and TNF-α). Interestingly, the cytokines modulated by the let-7 family are related to COVID-19-induced cytokine storm observed in patients undergoing clinical phase three. Thus, let-7 can be considered a novel and attractive biomarker for therapeutic purpose. Based on that, the present study aims to critically analyze the immunopathological mechanisms of the microRNA let-7 in the infection caused by SARS-CoV-2.

**Keywords:** COVID-19; SARS-CoV-2; cytokine storm; microRNA; let-7





## 1. Introduction

In December 2019, the previously unknown Severe Acute Respiratory Syndrome Coronavirus 2 (SARS-CoV-2) virus spread among the population of Wuhan, China. The rapid global advance of the virus and the thousands of deaths caused by coronavirus disease (COVID-19) have prompted the World Health Organization (WHO) to declare a state of pandemic on 12 March 2020 [1–3].

With a re-emerging pathogen such as the SARS-CoV-2, it remains absolutely necessary to gain a better understanding of the precise mechanisms of the SARS-CoV-2 pathophysiology and future viral variants [4]. In this context, the host immune response to the virus appears to play a key role in the disease pathogenesis and clinical presentation. An excessive inflammatory reaction, characterized by a marked pro-inflammatory cytokine release, is remarkable in patients with severe COVID-19. Thus, leading to immune abnormalities which may lead to persisted infections and septic shock [5–7].

Coronavirus has a 5′cap structure and 3′polyA tail. The spike glycoprotein (S), envelope (E), membrane (M) and nucleocapsid (N) are structural proteins in coronaviruses. S protein is cleaved to S1 and S2 subunits. S2 facilitates the entry into target cells [8–10]. A

major host protease, named transmembrane serine protease 2 (TMPRSS2), enables SARS-CoV-2 entry into host cells by priming the spike protein. SARS-CoV-2 uses the angiotensin converting enzyme 2 (ACE2) as a receptor for host cell attachment; once attached, the viral spike protein is cleaved by TMPRSS2 to allow fusion of the viral and cellular membranes [11,12]. The host protease TMPRSS2 plays a critical role in facilitating SARS-CoV-2 entry into host cells by priming the viral spike protein. Once the spike protein attaches to the host receptor ACE2, TMPRSS2 cleaves it, allowing fusion of the viral and cellular membranes. This process enables viral entry and subsequent replication, leading to the development of COVID-19. However, in some individuals, a dysregulated immune response occurs, resulting in a cytokine storm. This immune response involves the release of large amounts of pro-inflammatory cytokines and chemokines by immune effector cells, which can cause severe damage to host tissues and organs. The dysregulated immune response and cytokine storm are major causes of death in COVID-19. Therefore, TMPRSS2 may play a role not only in viral entry but also in the development of cytokine storm [13].

In a cytokine storm, the immune system produces an excessive amount of pro-inflammatory cytokines, such as interleukin-6 (IL-6), interleukin-1 beta (IL-1β), and tumor necrosis factor-alpha (TNF-α), among others. These cytokines can cause widespread inflammation throughout the body, which can lead to tissue damage and organ failure. In COVID-19, the cytokine storm is thought to contribute to the development of acute respiratory distress syndrome (ARDS), which is a severe lung condition that can lead to respiratory failure [14]. IL-6 is also involved in systemic inflammation during infection, which is probably influenced by pre-existing comorbidities [15]. This is followed by the infiltration of macrophages and neutrophils into the lung tissue, which results in a cytokine storm [16–18]. According to Jiang et al. (2022), further studies are still needed to fully elucidate the mechanism behind the cytokine storm induced by SARS-CoV-2 infection and thus provide new targets for therapeutic interventions [19].

The NF-κB signal transduction pathway is a common pathway centrally involved in the generation of pro-inflammatory cytokine and chemokine cascades in COVID-19 [20–22]. Inhibition of the NF-κB pathway increased survival rates in mice infected with SARS-CoV, due the fact that NF-κB signaling pathway activation is one of the major contributions to the inflammation induced upon SARS-CoV infection [23]. On a more molecular level, increased NF-κB activity leads to increased expression of Lin28, a gene that encodes an RNA-binding protein that plays a key role in regulating gene expression and developmental processes in many organisms, including humans. Lin28 has been shown to inhibit the maturation of let-7 family miRNAs, miRNAs that normally inhibits NF-κB transcripts and activators such as IL-6 [24].

MiRNAs can inhibit the viral translation after the attachment of miRNAs to 3′-UTR of the viral genome without affecting the expression of human genes [25,26]. In addition, the let-7 family is among the first microRNAs discovered. Most of the let-7 regulatory proteins are well studied in development, proliferation, differentiation, and cancer, but their role in inflammation and in infectious diseases such as COVID-19 is not well understood [27–29]. The present study aimed to systematically review the studies that were able to describe the role of let-7 in COVID-19, and to provide a basis for future research aimed at therapeutics in this area.

## 2. Materials and Methods

An extensive search was conducted in the MedLine database to identify studies addressing the relationship between let-7 family miRNAs and the pathophysiology of COVID-19. A keyword search was performed as follows: "COVID-19 or SARS-CoV-2" AND "let-7 or let7". The database was searched on 25 April 2022, for entries from 1 January 2020 to 24 April 2022. Publication format was limited to peer-reviewed journal articles (as a filter for quality resulting from the peer review process), including all types of publications. We included publications from any country in published in English language.

Subsequently, exclusion and inclusion criteria were applied. Studies that addressed the association between let-7 and the pathophysiology of COVID-19 were included, as well as studies performed in adults (older than 18 years). Studies that did not match the criteria described above were excluded (Figure 1).

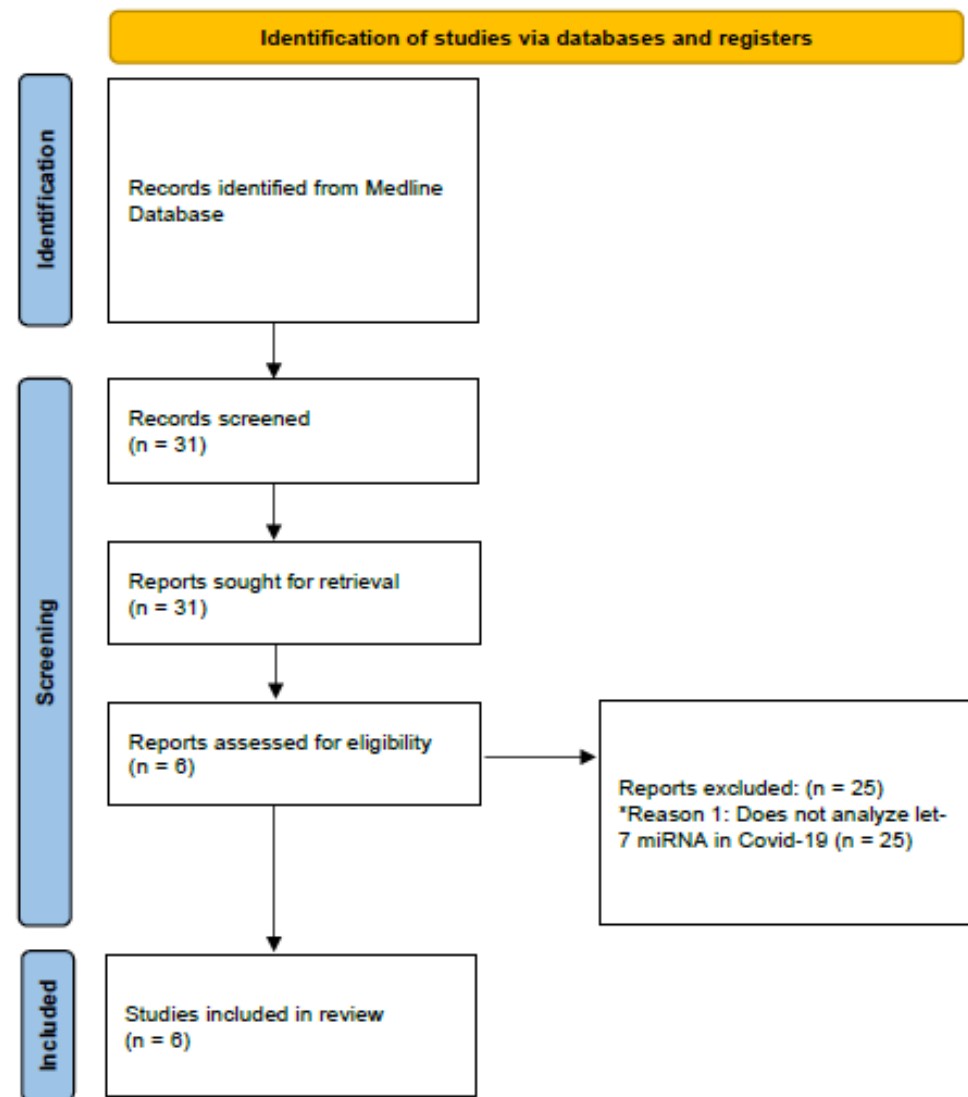

**Figure 1.** Diagram demonstrating the methodology used in the present review. * Reason 1: Reason for excluding articles.

*Risk of Bias*

The Risk of Bias in Non-randomized Studies–of Intervention (ROBINS-I) Tool was used to assess the methodological quality of all studies performed in human beings by evaluating the extent to which they addressed the possibility of bias in seven areas of study design [30]. Two researchers (RLP, GRA) independently assessed each included study and any uncertainty regarding the quality of publications was resolved through discussion among them.

**3. Results**

The search yielded 31 articles. After applying the search filter and considering the inclusion criteria defined for the present study, six studies capable of identifying the cellular mechanisms of let-7 in COVID-19 were selected. Table 1 describes the studies included in

the review. Table 2 presents a comparative summary of all the experiments included in the study.

**Table 1.** Articles included in the review results.

| Author | Journal | Year | Type of Survey |
|---|---|---|---|
| Nersisyan et al. [31] | PloS ONE | 2020 | Statistical analysis |
| Xie et al. [32] | Signal Transduct Target Ther | 2021 | In vitro |
| Chen et al. [33] | Front. Imunnol. | 2021 | In vivo |
| Wang Y. et al. [34] | Genome Research | 2022 | Case-control |
| Wang B. et al. [35] | Cell Death Discovery | 2022 | In vitro |
| Papannarao et al. [36] | Int J Obes | 2021 | Case-control |

**Table 2.** Characteristics and main findings of the analyzed articles.

| Author | Type of Survey | Population | Control | Interest | Outcome |
|---|---|---|---|---|---|
| Nersisyan et al. [31] | Statistical analysis | - | - | Revealing potential regulatory mechanisms of ACE2 and TMPRSS2 | Revealed strong indications that TMPRSS2 can be positively regulated by repressing hsa-let-7e transcription |
| Xie et al. [32] | In vitro | THP1 cells treated with pri-let-7 | Untreated THP1 cells | Hypothesis test that positive regulation of let-7 can attenuate the "cytokine storm" caused by SARS-CoV-2 | Cells overexpressing let-7a or let-7c reduced the mRNA level of IL-6 and significantly decreased the expression of many other cytokines and chemokines associated with COVID-19 |
| Chen et al. [33] | In vivo | Animal models of sepsis induced by cecal ligation and puncture (CLP) | (1) sham group, sham operation without treatment; (2) CLP group; (3) CLP + agomiR-NC group | Explore the downstream cytokines released by neutrophils following miR-let-7b treatment and its therapeutic effects | MiR-let-7b inhibited CLP-induced inflammation partially through the miR-let-7b/TLR4/NF-κB axis in neutrophils. Additionally, it significantly decreased IL-6 levels in mouse serum compared to the CLP group |
| Wang Y. et al. [34] | Case-control | 37 patients with COVID-19 (19 mild and 18 severe) | 8 healthy individuals | Characteristics of plasma cell-free RNA in the pathophysiology of COVID-19 | Low let-7 and high IL6 levels were observed in the COVID-19 patients |
| Wang B. et al. [35] | In vitro | WI-38 cells treated with cannabinoids | DMSO 0.025% | Revealing potential ACE2 and TMPRSS2 inhibitory mechanisms of cannabis extracts | It has been shown that some cannabis extracts can negatively regulate ACE2 and TMPRSS2 proteins by targeting miR-200c-3p and let-7a-5p |
| Papannarao et al. [36] | Case-control | 31 obese women | Thin individuals aged 30 years (±1.5) | Determine whether early changes in miRNAs are associated with ACE2 dysregulation | There was a negative correlation between miR-let-7b and ACE2 |

### 3.1. TMPRSS2

Studies have predicted that let-7 family miRNAs target the 3' UTR of the TMPRSS2 gene, a membrane protein that plays a crucial role during viral entry into the host cell, thus inhibiting its expression. Furthermore, it was reported that blocking the transcription of let-7 family members such as hsa-let-7e and hsa-let-7a-5p led to increased expression of TMPRSS2 gene [31,35].

### 3.2. NF-κB and TLR4

An in silico analysis of the TLR4 and NF-κB signaling pathways was performed and pointed out that miR-let-7b was among the top-ranked regulators of these inflammatory pathways, targeting 15 and 14 genes within the TLR4 and NF-κB pathways, respectively. Subsequently, animal models were used to investigate the role of miR-let-7b in inflammation in vivo. Animal models were induced to sepsis by cecal ligation and puncture. In sepsis-induced rat models, compared to control (sham-operation) the expression of nuclear TLR4 and NF-κB increased, while IκBα, a protein that participates in NF-κB inhibition, decreased. While in sepsis-induced mouse models group that received miR-let-7b-agomir treatment (10 nmol miR-let-7b-agomir per mouse suspended in 200 μL saline) was found to significantly decrease nuclear TLR4 and NF-κB and increased IκBα protein levels in neutrophils compared to those that did not receive the treatment. Furthermore, the group treated with miR-let-7b agomir significantly decreased the levels of IL-6 in mouse serum compared to the CLP group. Overall, these results demonstrated that miR-let-7b inhibited CLP-induced inflammation partially through the miR-let-7b/TLR4/NF-κB axis in neutrophils [33].

### 3.3. IL-6

A case–control experiment performed using 37 COVID-19 patients recruited from four local hospitals in Guangdong revealed increased IL-6 and decreased let-7 family miRNAs in patients of both the severe and mild groups compared to healthy subjects. The mild and severe groups also showed increased expression of the transcription factor NF-κB [34].

In this context, cocultivation of miR-let-7b mimetics with human neutrophils preconditioned by LPS was performed. Neutrophils can be activated by LPS to a pro-inflammatory state. Results with RT-qPCR and ELISA showed that after pre-treatment with LPS and incubation with miR-let-7b, neutrophils produced dramatically fewer pro-inflammatory cytokines, including IL-6, IL-8, and TNFα, and more anti-inflammatory cytokine IL-10 [33].

Additionally, studies have shown that TH1 cells play a crucial role in the immune response to COVID-19. TH1 cells are responsible for producing pro-inflammatory cytokines such as interferon-gamma (IFN-γ), which can help to clear the virus and prevent its spread. However, in severe cases of COVID-19, the immune response can become dysregulated, leading to a cytokine storm and potentially fatal complications. Let-7 miRNAs have been shown to regulate the expression of several pro-inflammatory cytokines, including IFN-γ, interleukin-6 (IL-6), and tumor necrosis factor-alpha (TNF-α). Overexpression of pri-let-7a and pri-let-7c in THP1 cells has been observed to reduce the mRNA levels of IL-6 and other cytokines and chemokines associated with SARS-CoV-2, such as IL-1β, IL-8, CCL2, GM-CSF, and TNFα. Conversely, a reduction in let-7 levels has been shown to increase the levels of these pro-inflammatory cytokines, suggesting that let-7 may play a crucial role in regulating the immune response to COVID-19 [32,37].

### 3.4. ACE2

An experiment conducted with 31 obese women in New Zealand showed a negative correlation between miR-let-7b and ACE2. Individuals with higher levels of expressed mir-let-7b exhibited lower levels of expressed ACE2 [36].

*3.5. Risk of Bias and Quality Assessment*

The overall quality of the studies was poor (Figure 2). The study by Wang Y. et al. had a moderate risk of bias regarding participant selection and confounding. The participant selection by Papannarao et al. had a serious risk of bias; the major factor for this assessment concerned the fact that the authors selected only female participants in the study design. Thus, of the two studies performed in humans included in our analysis, one was judged to have a moderate overall risk of bias, and one was judged to have a serious risk of bias [34,36].

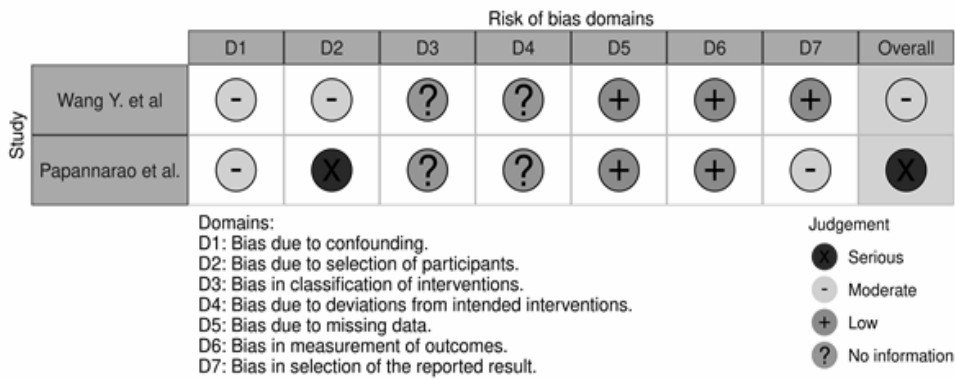

**Figure 2.** Risk of Bias assessment of the two clinical trials included in our analysis [34,36].

## 4. Discussion

MiRNA is a type of direct and potent regulator of gene expression. MiRNA controls gene expression by binding any regions suitable for interaction that can be located in DNA and RNA. The interplay between miRNA and other biomolecules is responsible for the homeostasis of a living organism [38]. Many aspects of the miRNA let-7 intersect with the immunopathology of COVID-19. The present review identified six experiments testing hypotheses about the role of let-7 in COVID-19 pathology, and these were critically summarized in the present article. Only two of these have been performed in humans. The Wang Y. et al. trial has the advantage of having dichotomous groups and recruiting patients from four different hospitals [34]. However, the low number of patients may influence the accuracy of the results.

Regarding the findings of each study, the most frequent finding was the increased expression of pro-inflammatory cytokines, especially IL-6, correlated with the suppression of let-7, generating the mechanism for the cytokine storm [33–35]. These findings are consistent with the literature of miRNA let-7 [38]. Most of the studies assessed were laboratory-based. They showed plausible correlations between aspects of COVID-19 immunopathology with let-7. Increased expression of TMPRSS2 via repression of let-7 (Figure 3) was cited by two authors [31,35]; this implies an increase in viral uptake into the target cells via two mechanisms: by the cleavage of SARS-S, which activates the S protein for membrane fusion, and by the cleavage of ACE2 [39,40]. The internalization and replication of virus subsequently causes degradation of membrane-bound ACE2 receptors, which in turn causes increase in angiotensin II and the angiotensin type 1 receptor, resulting in an inflammatory immune response [41]. Thus, decreased let-7 expression may play a role in increasing the viral load of the disease by facilitating virus entry into the host cells through increased expression of TMPRSS2.

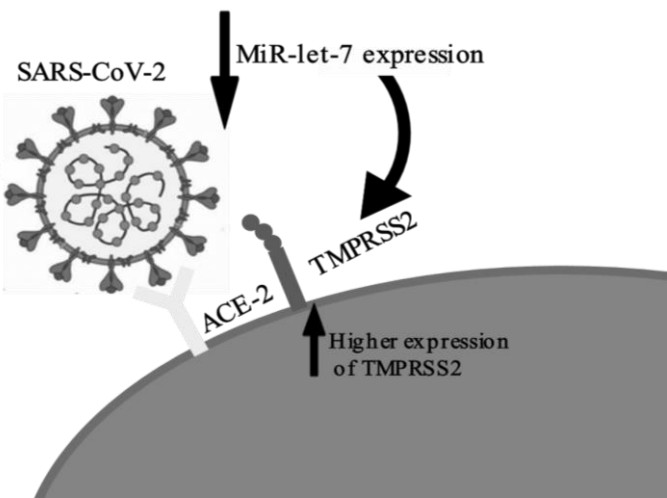

**Figure 3.** Representation of let-7 suppression by COVID-19 and the subsequent increase in TMPRSS2 expression.

The let-7 suppression also leads to increased activation of the TLR4/NF-κB pathway [33,42–44]. Activated NF-κB transcription factors are a trigger for the expression of a wide variety of cytokines, (e.g., IL-1, IL-2, IL-6 and TNF-α) that can induce an inflammatory programming in resident macrophages and recruit activated monocytes and T cells to the lungs [45,46]. In addition, one of the most highly induced NF-κB-dependent cytokines is IL-6; in patients with COVID-19, elevated IL-6 levels are associated with worse clinical outcomes [47]. Activation of NF-κB promotes the synthesis of the microRNA binding protein LIN-28, which reduces the synthesis of mature let-7, thus constituting an inflammatory loop involving NF-κB/IL-6/let-7/ACE2 [48,49]. These findings are plausible to explain the hyperinflammatory state and cytokine storm induced in severe cases of the disease, which results in tissue damage, acute respiratory distress (ARDS) and multiple organ failure [50,51].

Furthermore, overexpression of let-7 is able to suppress the hyperinflammatory state. Discovered by Xie C., C1632 is a small molecule that serves as a let-7 stimulator, capable of positively regulating let-7 and therefore reducing viral replication and the secretion of pro-inflammatory cytokines. This molecule had attractive results in in vitro studies as a potential therapeutic for COVID-19 [32]. Like C1632, other therapeutic targets may have therapeutic potential through the regulation of let-7, and this is an attractive biomarker for future therapeutic research.

The immunopathological mechanisms of COVID-19 are complex and pose a challenge to researchers. This study provides a critical analysis of let-7 miRNA in the context of COVID-19, serving as a basis for further research in the area. Our paper is the first comprehensive critical analysis published on let-7 and COVID-19. A limitation of our study was the low number of experiments performed, resulting in a low number of articles for analysis, and the low quality of studies performed in humans. As a strong point, our result has good immunopathological plausibility to explain clinical findings of the disease, and it provides reasonable data for future research in this area.

## 5. Conclusions

Six published experiments relating to let-7 and COVID-19 were identified. Some studies exhibited good accuracy for the correlation of let-7 in the immunopathology of COVID-19. The let-7 miRNA suppression generated in severe COVID-19 implies increased viral uptake by the target cell via increased TMPRSS2 and causes increased inflammation through the NF-κB/IL-6/let-7/LIN-28 axis. The future of let-7 targeted therapy is attractive, but there is a long way to go to obtain a clinically safe drug. Thus, future researchers may successfully set a direction in primary research by targeting let-7 as a therapeutic target.

**Author Contributions:** R.B.d.O., developed the idea, conceptualization, validation, writing (review and editing), supervision and project administration; R.L.P., performed the conceptualization, methodology, formal analysis, investigation, data curation and writing (original draft); G.d.R.A., performed methodology, formal analysis, investigation and data curation. All authors have read and agreed to the published version of the manuscript.

**Funding:** This research received no external funding.

**Institutional Review Board Statement:** Not applicable.

**Informed Consent Statement:** Not applicable.

**Data Availability Statement:** Not applicable.

**Acknowledgments:** We would like to thank Luiz Fernando Kehl for critically contribute to the article.

**Conflicts of Interest:** The authors declare no conflict of interest.

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
