# Peer review of "MicroRNA Let-7 Plays an Important Role in the Immunopathology of COVID-19: A Systematic Review"

_2673-5601, doi:10.3390/immuno3010008_

Round 1

Reviewer 1 Report

The authors reviewed six studies that examined and defined the functional roles of let-7 family miRNAs in regulating ACE/2TMPRSS2 expression as well as NFkB signaling axis. The manuscript has several English language errors that need to be rectified for a better meaning.

1. The authors indicated secreted for proteins LIN-28 and ACE2, both of which are not really secreted proteins. Correction required

2. Lines 110-112, please rephrase them for a better meaning

3. Lines 123-125, please clarify the mechanisms. Treatment with what?

4. Lines 143-145, rephrase for a better meaning

Author Response

Dear reviewer 1

Thanks for your corrections. All the changes requested were made and marked yellow in the text to facilitate the revision process. Once all remarks were related to direct corrections in the text, we opted to present a single response instead of a point-by-point one. Thanks in advance

Reviewer 2 Report

Dear Author, please find the minor comments

1) better to write re-emerging coronavirus 

2) Give the full form at first instance SARSCoV-2 -24th line

3) Line 30th - 35 - long, single sentence - please make three sentences.

4) Coronavirus mRNA will have Cap not coronavirus itself

5) Information in 43 line and 44 line is abrupt. Add connection there

6) Give full details on cytokine storm 

7) provide proper details on Lin 28

8) 64 th line - MiRNAs correct to miRNAs

9) Author mentioned - miRNA targeting proteins and receptor in text. please provide reference for that. As per my search I didn't find any such paper- miRNA binding to proteins. 

10) THP1 cells - add information on this in the manuscript.   

Author Response

Dear reviewer 2

Thanks for your corrections. All the changes requested were made and marked yellow in the text to facilitate the revision process. Once all remarks were related to direct corrections in the text, we opted to present a single response instead of a point-by-point one. Thanks in advance

Reviewer 3 Report

An interesting study on miRNA Let 7 which is associated with cytokine strom and could serve as a potential treatment target. Paper however needs some major revisions: 

Standard PRISMA flowchart to be used (https://guides.lib.unc.edu/prisma)

The studies in the results section should be eloborted better to give a better overview about the studies. 

Since COVID-19 is slowly waning off, adding a paragraph/couple of sentences in other viral illness or bacterial sepsis could be helpful as it could serve as potential target in other etiologies.  

Author Response

Dear reviewer 3

Thanks for your corrections. All the changes requested were made and marked yellow in the text to facilitate the revision process. Once all remarks were related to direct corrections in the text, we opted to present a single response instead of a point-by-point one. Thanks in advance